# Hospitalizations among adults with chronic kidney disease in the United States: A cohort study

Sarah J. Schrauben[1,2,3]*, Hsiang-Yu Chen[3], Eugene Lin[4], Christopher Jepson[5], Wei Yang[3], Julia J. Scialla[6], Michael J. Fischer[7,8], James P. Lash[8], Jeffrey C. Fink[9], L. Lee Hamm[10], Radhika Kanthety[11], Mahboob Rahman[12], Harold I. Feldman[1,2,3☉], Amanda H. Anderson[13☉], the CRIC Study Investigators[¶]

1 Renal, Electrolyte-Hypertension Division, Perelman School of Medicine at the University of Pennsylvania, Philadelphia, Pennsylvania, United States of America, 2 Center for Clinical Epidemiology and Biostatistics, Perelman School of Medicine at the University of Pennsylvania, Philadelphia, Pennsylvania, United States of America, 3 Department of Biostatistics, Epidemiology, and Informatics, Perelman School of Medicine at the University of Pennsylvania, Philadelphia, Pennsylvania, United States of America, 4 Department of Medicine, Division of Nephrology; Leonard D. Schaeffer Center for Health Policy & Economics, University of Southern California, Los Angeles, California, United States of America, 5 ECRI Institute, Plymouth Meeting, Pennsylvania, United States of America, 6 Department of Medicine, Division of Nephrology, University of Virginia School of Medicine, Charlotte, Virginia, United States of America, 7 Medical Service, Jesse Brown Veterans Affairs Medical Center, Chicago, Illinois, United States of America, 8 Department of Medicine, University of Illinois at Chicago, Chicago, Illinois, United States of America, 9 Department of Medicine, University of Maryland, Baltimore, Maryland, United States of America, 10 Tulane University School of Medicine, New Orleans, Louisiana, United States of America, 11 Department of Medicine, Case Western Reserve University, Cleveland, Ohio, United States of America, 12 Division of Nephrology and Hypertension, University Hospitals Cleveland Medical Center, Case Western Reserve University, Louis Stokes Cleveland Veterans Affairs Medical Center, Cleveland, Ohio, United States of America, 13 Department of Epidemiology, School of Public Health and Tropical Medicine, Tulane University, New Orleans, Louisiana, United States of America

☉ These authors contributed equally to this work.
¶ Membership of the CRIC Study Investigators is provided in the Acknowledgments.
* Sarah.Schrauben@pennmedicine.upenn.edu

**Data Availability Statement:** The data from the CRIC Study that support the findings of this study are available upon request at the NIDDK Repository at https://repository.niddk.nih.gov/studies/cric/ You

## Abstract

### Background

Adults with chronic kidney disease (CKD) are hospitalized more frequently than those without CKD, but the magnitude of this excess morbidity and the factors associated with hospitalizations are not well known.

### Methods and findings

Data from 3,939 participants enrolled in the Chronic Renal Insufficiency Cohort (CRIC) Study between 2003 and 2008 at 7 clinical centers in the United States were used to estimate primary causes of hospitalizations, hospitalization rates, and baseline participant factors associated with all-cause, cardiovascular, and non-cardiovascular hospitalizations during a median follow up of 9.6 years. Multivariable-adjusted Poisson regression was used to identify factors associated with hospitalization rates, including demographics, blood pressure, estimated glomerular filtration rate (eGFR), and proteinuria. Hospitalization rates in

must be a registered user on the NIDDK Repository website to access and use the data and sample request systems. You can register at https://repository.niddk.nih.gov/register/ Instructions for how to make a request are found at https://repository.niddk.nih.gov/pages/overall_instructions/ Please contact the NIDDK Repository Communications team (niddk-cr@imsweb.com) should you have any questions about the requests or are unsure of which one fits your needs. The data from the National Inpatient Sample (NIS) that support the findings of this study are available upon request and purchase from the Healthcare Cost and Utilization Project (HCUP). Restrictions apply to the availability of these data, which were used under license for this study. Data are available for purchase online through the Online HCUP Central Distribtuion with permission of HCUP at: https://www.hcup-us.ahrq.gov/tech_assist/centdist.jsp Questions about purchasing databases can be directed to the HCUP Central Distributor: Email: HCUPDistributor@AHRQ.gov Telephone: (866) 556-4287 (toll free) Fax: (866) 792-5313 (toll free).

**Funding:** This work was supported in part by the National Institutes of Health (NIH) through the National Institute of Diabetes and Digestive and Kidney Diseases (NIDDK), https://www.niddk.nih.gov: SJS receives support from NIDDK K23DK 118198-01A1 and EL receives support from NIDDK K08DK118213. EL also receives support from the University Kidney Research Organization (https://ukrocharity.org). The content is solely the responsibility of the authors and does not necessarily represent the official views of the National Institutes of Health. Funding for the CRIC Study was obtained under a cooperative agreement from NIDDK to the CRIC Study Investigators (U01DK060990, U01DK060984, U01DK061022, U01DK061021, U01DK061028, U01DK060980, U01DK060963, U01DK060902 and U24DK060990). In addition, this work was supported in part by several NIH Clinical and Translational Science Awards (NCATS) to the CRIC Study Investigators; https://ncats.nih.gov: the Perelman School of Medicine at the University of Pennsylvania NIH/NCATS UL1TR000003, Johns Hopkins University UL1 TR-000424, University of Maryland GCRC M01 RR-16500, Clinical and Translational Science Collaborative of Cleveland, UL1TR000439 from the NCATS, Michigan Institute for Clinical and Health Research (MICHR) UL1TR000433, University of Illinois at Chicago CTSA UL1RR029879, Tulane COBRE for Clinical and Translational Research in Cardiometabolic Diseases P20 GM109036, Kaiser Permanente NIH/NCRR UCSF-CTSI UL1 RR-024131, Department of

CRIC were compared with rates in the Nationwide Inpatient Sample (NIS) from 2012. Of the 3,939 CRIC participants, 45.1% were female, and 41.9% identified as non-Hispanic black, with a mean age of 57.7 years, and the mean eGFR is 44.9 ml/min/1.73m$^2$. CRIC participants had an unadjusted overall hospitalization rate of 35.0 per 100 person-years (PY) [95% CI: 34.3 to 35.6] and 11.1 per 100 PY [95% CI: 10.8 to 11.5] for cardiovascular-related causes. All-cause, non-cardiovascular, and cardiovascular hospitalizations were associated with older age (≥65 versus 45 to 64 years), more proteinuria (≥150 to <500 versus <150 mg/g), higher systolic blood pressure (≥140 versus 120 to <130 mmHg), diabetes (versus no diabetes), and lower eGFR (<60 versus ≥60 ml/min/1.73m$^2$). Non-Hispanic black (versus non-Hispanic white) race/ethnicity was associated with higher risk for cardiovascular hospitalization [rate ratio (RR) 1.25, 95% CI: 1.16 to 1.35, *p*-value < 0.001], while risk among females was lower [RR 0.89, 95% CI: 0.83 to 0.96, *p*-value = 0.002]. Rates of cardiovascular hospitalizations were higher among those with ≥500 mg/g of proteinuria irrespective of eGFR. The most common causes of hospitalization were related to cardiovascular (31.8%), genitourinary (8.7%), digestive (8.3%), endocrine, nutritional or metabolic (8.3%), and respiratory (6.7%) causes. Hospitalization rates were higher in CRIC than the NIS, except for non-cardiovascular hospitalizations among individuals aged >65 years. Limitations of the study include possible misclassification by diagnostic codes, residual confounding, and potential bias from healthy volunteer effect due to its observational nature.

## Conclusions

In this study, we observed that adults with CKD had a higher hospitalization rate than the general population that is hospitalized, and even moderate reductions in kidney function were associated with elevated rates of hospitalization. Causes of hospitalization were predominantly related to cardiovascular disease, but other causes contributed, particularly, genitourinary, digestive, and endocrine, nutritional, and metabolic illnesses. High levels of proteinuria were observed to have the largest association with hospitalizations across a wide range of kidney function levels.

## Author summary

### Why was this study done?

- Chronic kidney disease (CKD) (or non-dialysis-dependent kidney disease) is increasingly common globally, and individuals with CKD have a high risk of health complications, including hospitalizations.

- Many of the hospitalizations experienced by those with CKD are thought to be due to cardiovascular disease, but little else is known about the other causes for hospitalization or why people with kidney disease are at higher risk of hospitalizations.

- Learning more about causes of hospitalization and risk factors for hospitalizations can guide outpatient management.

- Research to date on hospitalizations in kidney disease has mainly focused on those with dialysis-dependent kidney disease.

Internal Medicine, University of New Mexico School of Medicine Albuquerque, NM R01DK119199. The funders had no role in study design, data collection and analysis, interpretation of results, decision to publish, or preparation of the manuscript.

**Competing interests:** I have read the journal's policy and the authors of this manuscript have the following competing interest: EL receives consulting income from Acumen, LLC.

**Abbreviations:** AHRQ, Agency for Healthcare Research and Quality; CCS, Clinical Classification System; CKD, chronic kidney disease; CRIC, Chronic Renal Insufficiency Cohort; eGFR, estimated glomerular filtration rate; ESKD, end-stage kidney disease; GI, gastrointestinal; GU, genitourinary; ICD-9, International Classification of Diseases, ninth revision; KDIGO, Kidney Disease Improving Global Outcomes; NH, non-Hispanic; NIS, Nationwide Inpatient Sample; Y, person-years; RR, rate ratio; SBP, systolic blood pressure; STROBE, Strengthening the Reporting of Observational Studies in Epidemiology; UPCR, urine protein-to-creatinine ratio.

## What did the researchers do and find?

- We looked at hospitalization data from adults with CKD who were followed for nearly 10 years.

- We classified hospitalizations by the primary discharge code and found that non-cardio-vascular causes, such as genitourinary-, digestive-, and endocrine-related causes, comprised the majority of hospitalizations and that the largest single contributor to hospitalizations was due to cardiovascular reasons.

- We modeled the risk of hospitalizations with patient characteristics, such as age and sex, and clinical factors, such as level of kidney function, blood pressure, and proteinuria, and found that high levels of proteinuria were found to have a high risk of hospitalization regardless of kidney function level, and the risk of hospitalizations occurred even at moderate levels of kidney function.

- We also compared the hospitalization data from individuals with known CKD to a sample of the general hospitalized population in the United States and found that adults with CKD have higher hospitalization rates than this general sample.

## What do these findings mean?

- These findings highlight the need for developing better approaches to identifying patients at risk for severe complications of CKD and to guiding outpatient management strategies to improve outcomes in CKD.

- The findings may be particularly relevant to health care providers in general medicine since the increased risk of hospitalization occurred with even moderate reductions in kidney function, which do not typically correspond to being under the care of a kidney disease expert.

- Our study's findings might not be applicable to other CKD populations since the study enrolled volunteers, who might be healthier than other populations.

## Introduction

The prevalence of chronic kidney disease (CKD) is high, affecting up to 15% to 20% of the adult population in the United States (US) [1–3]. Numerous studies have provided evidence that declining kidney function is associated with increased hospitalization, with CKD as an important predictor of hospitalizations, even after accounting for comorbidities [3–8]. Hospitalization rates for individuals with CKD have been reported to be more than twice the rate of those without CKD [9]. Hospitalizations that occur in people with kidney disease are also more likely to lead to further complications, including greater re-hospitalization rates, higher mortality, longer lengths of stay, and worsening of kidney function, compared with patients without CKD [10–13]. However, incomplete information on comorbidities and inadequate study of underrepresented populations within the broader US CKD population limit inferences from these studies regarding the burden of hospitalization in the setting of CKD. As well, prior published reports in this population do not permit extensive characterization of the risk factors for hospitalization [4,11,14–18].

Cardiovascular disease has been identified as a major cause of hospitalization among those with advanced kidney disease, but little is known about the other causes contributing to the high burden of hospitalization among those with earlier stages of CKD [3,6,19,20]. Additionally, prior research on hospitalizations in populations with kidney disease has been limited by studying only 1 center [6], 1 region [10], certain age groups [11,14], 1 sex [15], or only those with end-stage kidney disease (ESKD) [7]. As well, prior studies have primarily used retrospective claims data [6,7,10,11,21] and cross-sectional data [4,16–18] or focused primarily on healthcare costs, without accounting for proteinuria [8,22–24]. Further research is needed to characterize the major causes of hospitalization in CKD, which could guide more intensive outpatient management strategies to reduce hospitalization and ultimately reduce the morbidity and healthcare costs in this high-risk population.

In this manuscript, we characterize the burden of hospitalizations within a diverse study population with mild-to-moderate CKD enrolled in the multicenter observational Chronic Renal Insufficiency Cohort (CRIC) Study. We also explore associations of demographic and kidney-specific factors with rates of hospitalization, characterize the leading causes of hospitalization, and compare the rates of hospitalizations in this population with rates in a sample representative of nearly all hospitalizations in the US general population.

## Methods

This study is reported as per the Strengthening the Reporting of Observational Studies in Epidemiology (STROBE) guideline (S1 Table).

### Study design and study populations

The CRIC Study enrolled a total of 3,939 men and women between 2003 and 2008 at 7 clinical centers (University of Pennsylvania, Johns Hopkins University, Case Western Reserve University, University of Michigan, University of Illinois at Chicago, Tulane University, and Kaiser Permanent of Northern California) across the US. Eligibility criteria have been previously described [25,26]. Age-specific estimated glomerular filtration rate (eGFR) criteria for eligibility were as follows: 20 to 70 mL/min/1.73m$^2$ for individuals aged 21 to 44 years, 20 to 60 mL/min/1.73m$^2$ for individuals aged 45 to 64 years, and 20 to 50 mL/min/1.73m$^2$ for individuals aged 65 to 74 years. Major exclusion criteria included prior dialysis lasting more than 1 month, NYHA Class III/IV heart failure, polycystic kidney disease, or other primary renal diseases requiring active immunosuppression. Participants completed annual clinic visits at which data were obtained, and blood and urine specimens were collected.

The National Inpatient Sample (NIS) was utilized as a representation of US nationwide hospitalizations from the year 2012. The NIS is the largest available all-payer inpatient database in the public domain that includes hospital discharge data, reflecting approximately 95% of all hospital discharges within the US [27]. The year 2012 was meant to be illustrative in comparison with the CRIC cohort and was chosen because it represented a year when the entire original CRIC cohort was enrolled and hospitalization ascertainment was optimized. The NIS provides discharge-level demographic and clinical characteristics that are searchable using the International Classification of Diseases, ninth revision (ICD-9) or the Clinical Classification System (CCS) codes and is sponsored by the Agency for Healthcare Research and Quality (AHRQ) and the Healthcare Cost and Utilization Project [28].

### Study variables

**Outcomes.** Hospitalizations between 2003 and 2018 among CRIC participants were ascertained through self-report and hospital queries and confirmed after review of medical records

by study personnel. Any hospitalization that occurred during the follow-up period was accounted for. The unit of observation in the NIS was an inpatient stay record, and after applying discharge weights, the amount of discharges in the US was estimated [29].

The length of hospitalization was calculated as the date of discharge minus the date of admission, plus one. Hospitalizations that were longer than 1 calendar day are the primary focus of this paper; hospital stays with an admission and discharge on the same calendar day are classified as ≤1-day hospitalizations. To characterize the specific cause of each hospitalization, the primary ICD-9 or ICD-10 admission code was extracted from the hospitalization discharge record and classified into 1 of 18 categories using the CCS Software developed by the AHRQ [28]. The cause of each hospitalization was further classified as "cardiovascular" if it met the AHRQ-defined categorization of "diseases of the circulatory system," such as hypertension, coronary atherosclerosis and other heart disease, valvular and conduction disorders, heart failure, cerebrovascular disease, and peripheral vascular disease, and if the primary cause did not meet the definition, the hospitalization was classified as "non-cardiovascular." Please see S1 Text for further details regarding the CCS categories.

**Covariates.** All considered covariates for the CRIC Study were ascertained at the baseline study visit. We examined age, sex, education, race/ethnicity, diabetes status (based on a fasting glucose of ≥126 mg/dL, a non-fasting glucose of ≥200 mg/dL, or self-reported use of insulin or other medications for glycemic control), systolic blood pressure (SBP) calculated as an average of 3 standardized measurements, eGFR assessed with a CRIC-specific equation using serum creatinine and cystatin C (ml/min/1.73m$^2$) [30], and urine protein-to-creatinine ratio (UPCR), calculated as urine total protein concentration divided by urine creatinine concentration (mg/g). The site of recruitment was also included to account for patterns of hospitalization potentially related to specific geographic regions and case mix. In the NIS data, age was ascertained from the discharge record, and the distribution of diabetes and race/ethnicity (black, white, Hispanic, and other) was estimated from US national prevalence reports [31,32].

## Statistical analysis

The analyses closely followed the original planning document. Please see S2 Text for further details.

The CRIC Study population was described using means, standard deviations, percentages, medians, and interquartile ranges; differences in key characteristics across categories of eGFR were assessed using ANOVA, chi-squared test, and Kruskal–Wallis test as appropriate. The follow-up time used in the analyses included time until the onset of ESKD, withdrawal, death, or end of the current follow-up period (early 2018), whichever occurred first, for median follow-up of 9.6 years.

Unadjusted rates of all-cause, cardiovascular-related, and non-cardiovascular-related hospitalizations were calculated in the CRIC Study as the total number of hospitalizations that occurred during follow-up divided by the total number of observed person-years (PY) outside of the hospital for all participants and stratified by baseline demographic and kidney-specific characteristics (age, sex, race/ethnicity, education level, diabetes status, and categories of SBP, eGFR, and UPCR).

Hospitalization rates were adjusted for age, race, and diabetes using Poisson regression models with the length of follow-up included as an offset term to account for varying durations of follow-up. Models examined the multivariable-adjusted associations of key baseline demographic and kidney-specific covariates (age <45, 45 to 64, >64 years), race/ethnicity (non-Hispanic [NH] white, NH black, Hispanic, other), sex (male, female), education level (less than

high school, high school, some college, college graduate or higher), clinical center (University of Pennsylvania, Johns Hopkins University, Case Western Reserve University, University of Michigan, University of Illinois at Chicago, Tulane University, and Kaiser Permanent of Northern California), diabetes (no, yes), SBP (<120, 120 to <130, 130 to <140, ≥140 mmHg), eGFR (<30, 30 to <45, 45 to <60, ≥60 ml/min/1.73m$^2$), and UPCR (<150, 150 to <500, ≥500 mg/g) with hospitalization rates. We also modeled age, SBP, eGFR, and UPCR as continuous variables as per unit change in response to reviewers. Poisson regression modeling was utilized to estimate a relative risk, reported as rate ratios (RR) and 95% confidence intervals (CI). In addition to main effects, we explored an interaction between proteinuria (UPCR < 150, 150 to <500, ≥500 mg/g) and eGFR (<30, 30 to <45, 45 to <60, ≥60 ml/min/1.73m$^2$).

The frequency of each primary cause of hospitalization (using the 18 categories of primary cause generated by the CCS Software) during the follow-up period was calculated overall and by diabetes status. We compared the hospitalization rates (all-cause, cardiovascular-related, non-cardiovascular-related) among participants in the CRIC Study and the rates among those in the NIS sample within 4 age group categories (≤45 years, 46 to 55 years, 56 to 65 years, ≥66 years), adjusted for the national distribution of race and diabetes. In both data sets, the rate was ascertained by summing hospitalizations for a given age group over the person-time of the follow-up in that group. Hospitalization rates in the NIS were calculated by first applying discharge weights to the NIS discharge data to estimate the number of hospital discharges that occurred in the US in 2012 [29] and then divided this number by the US population determined by the census.

### Ethics statement

The CRIC Study protocol was approved by the institutional review boards of all participating centers: the University of Pennsylvania Institutional Review Board (Federalwide Assurance # 00004028), Johns Hopkins Institutional Review Board (Study # NA_00044034 / CIR00004697), The University of Maryland, Baltimore Institutional Review Board, University Hospitals Cleveland Medical Center Institutional Review Board, MetroHealth Institutional Review Board, Cleveland Clinic Foundation Institutional Review Board (IRB #5969), University of Michigan Medical School Institutional Review Board (IRBMED), Wayne State University Institutional Review Board, University of Illinois at Chicago Institutional Review Board, Tulane Human Research Protection Office, Institutional Review Boards, Biomedical Social Behavioral (reference#: 140987), and Kaiser Permanente Northern California Institutional Review Board. The study is in accordance with the Declaration of Helsinki. All CRIC participants provided written informed consent.

### Results

The analysis included data from 3,939 CRIC Study participants (mean age 57.7 years; 54.9% male). Baseline characteristics of the study population are reported overall and by eGFR category (Table 1). Age, sex, race, education, UPCR and SBP levels, and diabetes significantly differed across eGFR categories. A total of 11,603 hospitalizations occurred between 2003 and 2018, for a median follow-up of 9.6 (interquartile range [IQR] 4.2 to 12.5) years.

### All-cause, cardiovascular, and non-cardiovascular hospitalization rates

The unadjusted all-cause hospitalization rates were 35.0 per 100 PY (95% CI: 34.3 to 35.6), with non-cardiovascular rates 23.9 per 100 PY (95% CI: 23.3 to 24.4) and cardiovascular-specific rates less than one-third of the all-cause rates (11.1 per 100 PY, 95% CI: 10.8 to 11.5) (S2

**Table 1. Characteristics of CRIC participants at baseline overall and by eGFR (*N* = 3,939).**

| Characteristic | Overall (*N* = 3,939) | By eGFR (ml/min/1.73m$^2$) | | | | |
|---|---|---|---|---|---|---|
| | | <30 (*N* = 806) | 30 to <45 (*N* = 1,340) | 45 to <60 (*N* = 1,091) | ≥60 (*N* = 702) | *p*-value* |
| Age, years | 57.7 (11) | 58.4 (10.9) | 59.2 (10.9) | 58.2 (10.6) | 52.9 (10.6) | <0.001 |
| Sex | | | | | | <0.001 |
| Male | 2,161 (54.9%) | 402 (49.9%) | 712 (53.1%) | 650 (59.6%) | 397 (56.6%) | |
| Female | 1,778 (45.1%) | 404 (50.1%) | 628 (46.9%) | 441 (40.4%) | 305 (43.4%) | |
| Race/ethnicity | | | | | | <0.001 |
| Non-Hispanic white | 1,638 (41.6%) | 255 (31.6%) | 529 (39.5%) | 490 (44.9%) | 364 (51.9%) | |
| Non-Hispanic black | 1,650 (41.9%) | 370 (45.9%) | 583 (43.5%) | 439 (40.2%) | 258 (36.8%) | |
| Hispanic | 497 (12.6%) | 153 (19%) | 186 (13.9%) | 117 (10.7%) | 41 (5.8%) | |
| Other | 154 (3.9%) | 28 (3.5%) | 42 (3.1%) | 45 (4.1%) | 39 (5.6%) | |
| Education | | | | | | <0.001 |
| Less than high school | 828 (21%) | 250 (31%) | 341 (25.4%) | 184 (16.9%) | 53 (7.6%) | |
| High school graduate | 741 (18.8%) | 175 (21.7%) | 265 (19.8%) | 202 (18.5%) | 99 (14.1%) | |
| Some college | 1,146 (29.1%) | 220 (27.3%) | 383 (28.6%) | 341 (31.3%) | 202 (28.8%) | |
| College graduate or higher | 1,223 (31.1%) | 161 (20%) | 351 (26.2%) | 364 (33.4%) | 347 (49.5%) | |
| Diabetes | 1,908 (48.4%) | 479 (59.4%) | 737 (55%) | 492 (45.1%) | 200 (28.5%) | <0.001 |
| Systolic blood pressure (mmHg) | 129 (22) | 133 (24) | 130 (23) | 127 (21) | 122 (19) | <0.001 |
| Urine protein-creatinine ratio (mg/g) | 152.6 (57.5–777.5) | 675.9 (138.3–2295.4) | 236.2 (68.4–1096.5) | 100.6 (50.6–364.3) | 64.1 (41.6–135.4) | <0.001 |
| eGFR (ml/min/1.73m$^2$) | 44.9 (16.7) | 24.3 (3.8) | 37.5 (4.2) | 51.9 (4.4) | 71.7 (10.3) | <0.001 |

For categorical variables, reported as number (%, percentage), and for continuous variables, mean and standard deviation is reported if normally distributed and mean and interquartile range if not normally distributed.

CRIC, Chronic Renal Insufficiency Cohort; eGFR, estimated glomerular filtration rate.

*$\chi^2$ test for categorical variables and the ANOVA test or the Kruskal–Wallis test for continuous variables.

Table). Age-, race-, and diabetes-adjusted all-cause, cardiovascular, and non-cardiovascular-related hospitalization rates were highest among older participants (≥65 years), NH black participants, and those with diabetes (see Fig 1 and S3 Table).

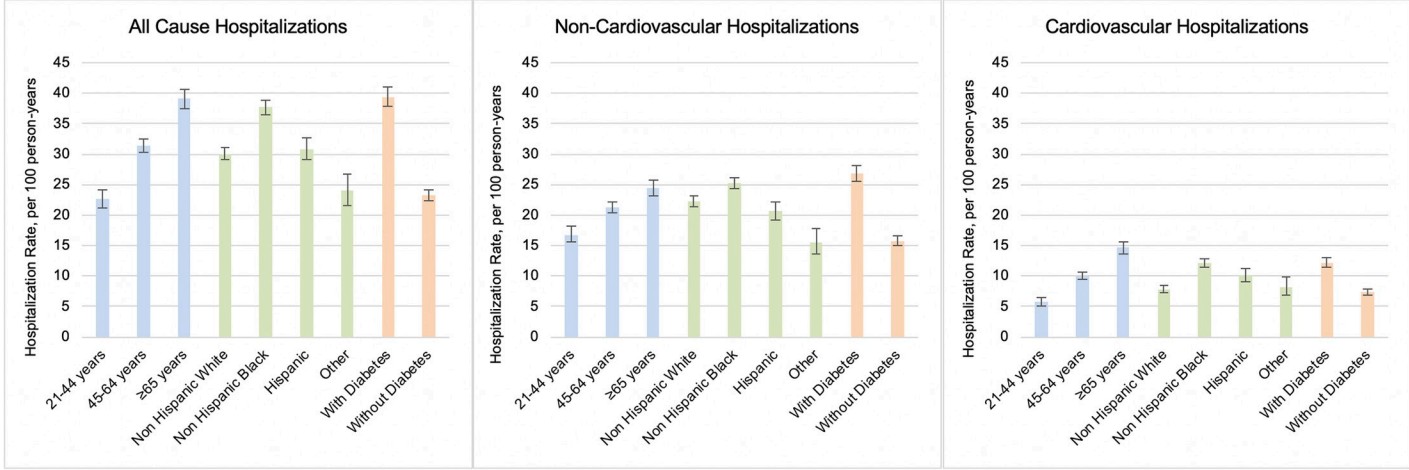

**Fig 1. Age-, race-, and diabetes-adjusted rates for all-cause, non-cardiovascular, and cardiovascular hospitalizations in CRIC participants (*N* = 3,939).** Rates reported as per 100 PY. Error bars indicate 95% confidence intervals. CRIC, Chronic Renal Insufficiency Cohort; PY, person-years.

## Association of baseline demographic and kidney-specific characteristics and hospitalization rates

Multivariable-adjusted associations of demographic and kidney-related characteristics and hospitalization rates, reported as RR and 95% CI using categorical versions of age, SBP, UPCR, and eGFR are shown in Fig 2 and S4 Table. When modeled continuously, these variables demonstrated a linear relationship with hospitalization rates (S1 Fig). The adjusted associations of non-cardiovascular hospitalizations largely aligned with those of all-cause hospitalizations. Age ≥65 years (versus 45 to 64 years), diabetes (versus no diabetes), SBP ≥140 mmHg (versus SBP 120 to <130 mmHg), UPCR categories ≥500 and 150 to <500 mg/g (versus UPCR <150 mg/g), and an eGFR categories <30, 30 to <45, 45 to <60 ml/min/1.73m$^2$ (versus eGFR ≥60 ml/min/1.73m$^2$), particularly at an eGFR <45 ml/min/1.73m$^2$, were associated with higher rates of hospitalizations for all-causes, cardiovascular, and non-cardiovascular causes (please see Fig 2 and S4 Table). Adjusted associations of baseline characteristics with hospitalization rate ratios were similar in magnitude and direction across types (all-cause, cardiovascular,

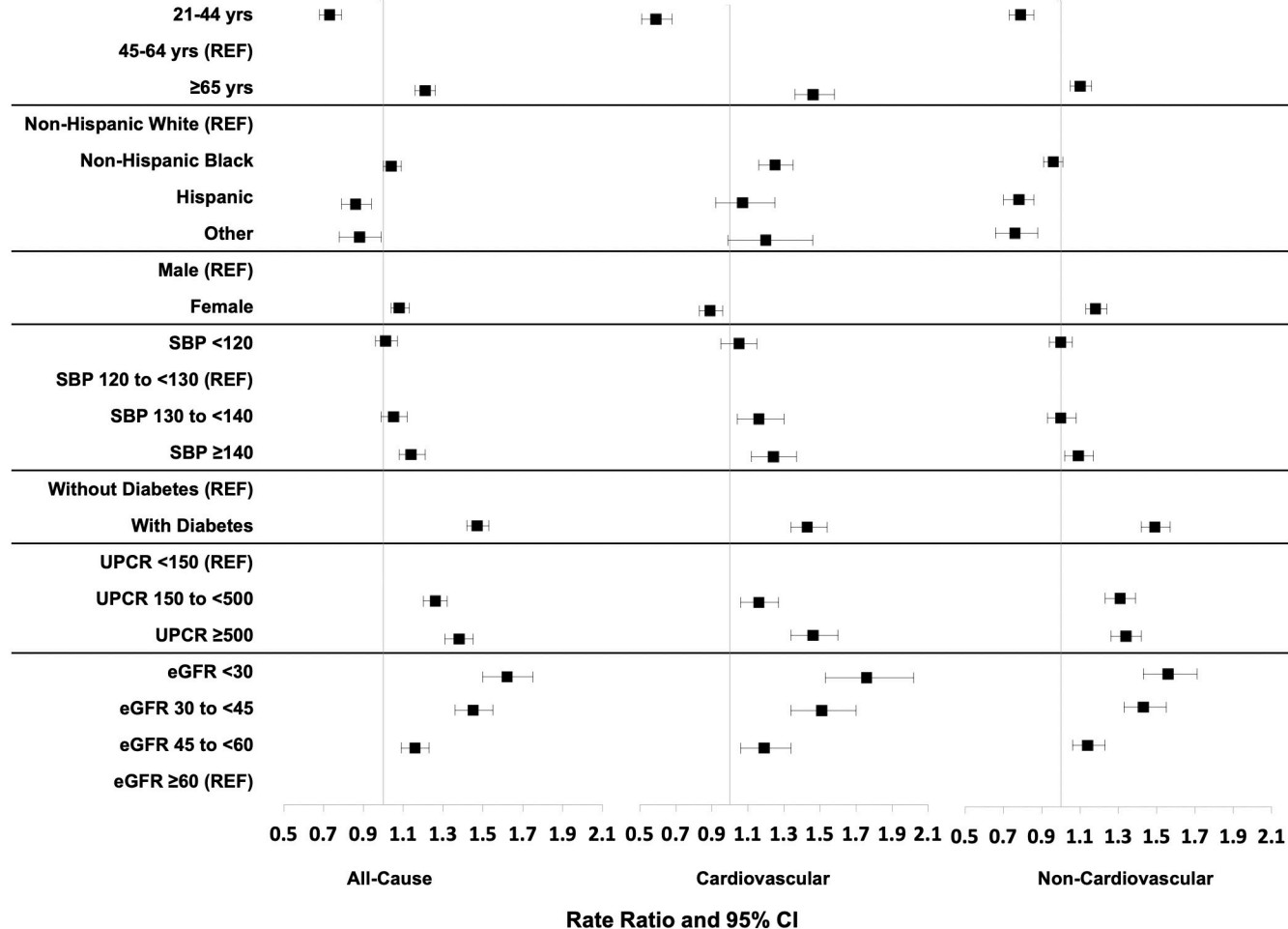

**Fig 2. Plot of adjusted all-cause, cardiovascular, and non-cardiovascular hospitalization rate ratios.** Error bars indicate 95% confidence intervals. Models adjusted for age, race, sex, clinical center, education, SBP, diabetes, UPCR, and CRIC baseline eGFR. CRIC, Chronic Renal Insufficiency Cohort; eGFR, estimated glomerular filtration rate; SBP, systolic blood pressure; UPCR, urine protein-to-creatinine ratio.

non-cardiovascular), except for sex, race/ethnicity, and SBP. Compared with males, female participants experienced higher rates of all-cause and non-cardiovascular hospitalizations, but lower rates of cardiovascular hospitalizations (all-cause: RR 1.08, 95% CI: 1.04 to 1.13, *p*-value < 0.001; non-cardiovascular: RR 1.18, 95% CI: 1.13 to 1.24, *p*-value < 0.001; cardiovascular: RR 0.89, 95% CI: 0.83 to 0.96, *p*-value = 0.002). Compared with NH whites, NH black participants experienced significantly higher cardiovascular hospitalization rates, but almost the same all-cause and non-cardiovascular hospitalizations (cardiovascular: RR 1.25, 95% CI: 1.16 to 1.35, *p*-value < 0.001; all-cause: RR 1.04, 95% CI: 1.00 to 1.09, *p*-value < 0.001; non-cardiovascular: RR 0.96, 95% CI: 0.91 to 1.01, *p*-value < 0.001). An SBP level of ≥140 mmHg (versus 120 to <130) was associated with more all-cause and non-cardiovascular hospitalizations (all-cause: RR 1.14, 95% CI: 1.08 to 1.21, *p*-value < 0.001; non-cardiovascular: RR 1.09, 95% CI: 1.02 to 1.17, *p*-value = 0.017), but an SBP level of 130 to <140 was associated with increased cardiovascular hospitalizations (RR 1.16, 95% CI: 1.04 to 1.30, *p*-value < 0.001). There was a consistent dose-dependent increase on all types of hospitalization as proteinuria increased and as eGFR decreased (Fig 2 and S4 Table).

An interaction between eGFR and proteinuria was detectable for all types of hospitalizations (*p*-value < 0.001) (Fig 3 and S5 Table). The observed effect of each factor on hospitalization rates was largely restricted to participants with relatively preserved kidney function as measured by the other factor. Participants with an eGFR <30 ml/min/1.73m$^2$ experienced high hospitalization rates across UPCR categories of <150 mg/g and ≥500 mg/g and participants with moderate-heavy proteinuria (UPCR of ≥500 and 150 to <500 mg/g) experienced high hospitalization rates across eGFR levels.

## Primary causes of hospitalizations

The 5 most common causes of hospitalization were cardiovascular-related (31.8%); genitourinary (GU)-related (8.7%); digestive/gastrointestinal (GI)-related (8.3%); endocrine, nutritional, or metabolic and immunity-related (8.3%); and respiratory-related (6.7%) (Fig 4). The patterns among participants with diabetes mirror the overall pattern just described. Participants without diabetes experienced fewer admissions due to endocrine, nutritional, or metabolic and immune causes (4.2% versus 11.6%) and more admissions due to musculoskeletal/connective tissue causes (8.1 versus 4.9%) (S2 Fig).

## Age-specific hospitalization rates in the CRIC study and national inpatient sample

All-cause adjusted hospitalization rates were observed to be higher in the CRIC cohort than the NIS for each age category (≤45 years: 19.1 versus 7.7; 46 to ≤55 years: 24.1 versus 9.3; 56 to ≤65 years: 26.3 versus 13.1; >65 years: 32.3 versus 31.5 per 100 PY) (Fig 5). Cardiovascular hospitalization rates were also higher in the CRIC cohort than in the NIS across each age category (≤45 years: 4.1 versus 0.2; 46 to ≤55 years: 6.9 versus 1.6; 56 to ≤65 years: 7.8 versus 2.9; >65 years: 11.3 versus 8.2 per 100 PY). Non-cardiovascular hospitalization rates remained higher in the CRIC Study for individuals aged ≤65 years (≤45 years: 15.0 versus 7.4; 46 to ≤55 years: 17.2 versus 7.6; 56 to ≤65 years: 18.4 versus 10.3) and were similar to the NIS for those aged >65 years (20.8 versus 23.3 per 100 PY). In general, discrepancies between hospitalization rates in the CRIC and NIS samples were greatest in the younger age groups; for individuals over 65 years of age, all-cause hospitalization rates in the 2 samples were more similar.

Unadjusted and adjusted hospitalization rates of hospital stays ≤1-day are displayed in S6–S9 Tables and S3 Fig.

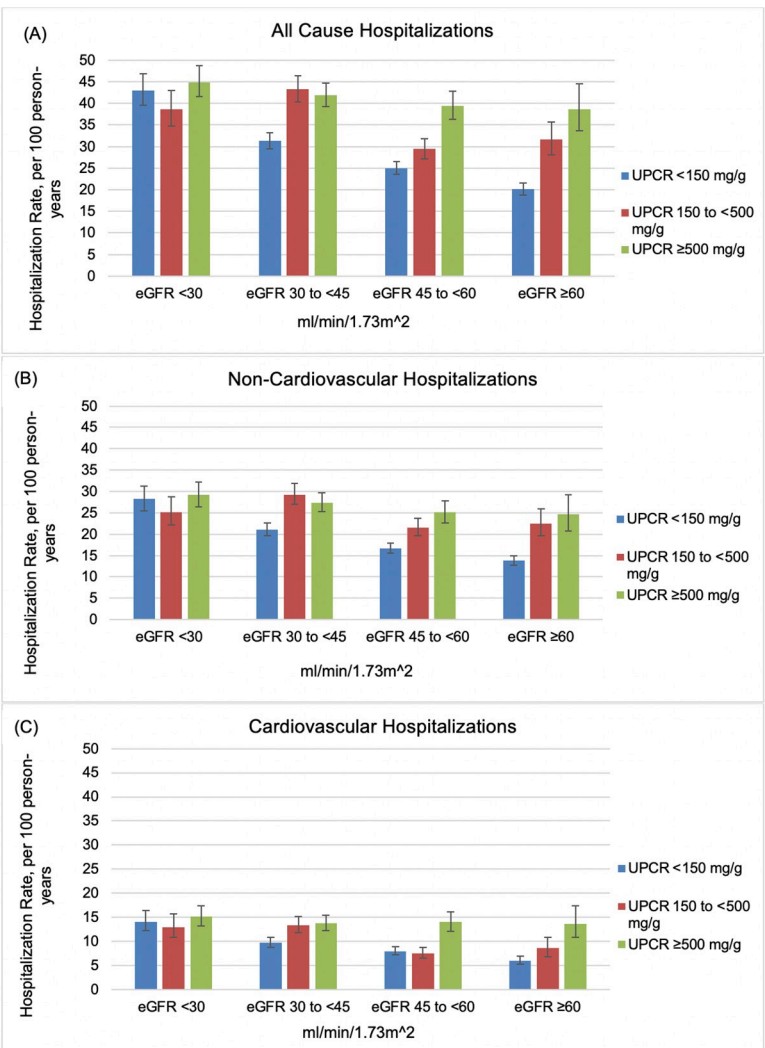

**Fig 3. Adjusted rates of hospitalization for (A) all-causes, (B) non-cardiovascular, and (C) cardiovascular, by level of eGFR and proteinuria.** Models adjusted for age, race, sex, clinical center, education, SBP, diabetes, UPCR, eGFR, and interaction between UCPR and eGFR. eGFR, estimated glomerular filtration rate; SBP, systolic blood pressure; UPCR, urine protein-to-creatinine ratio.

## Discussion

In the current investigation of CRIC Study participants, a well-characterized cohort of individuals with mild-to-moderate CKD, we observed that while the most common reason for hospitalization was related to cardiovascular disease, the majority of hospitalizations were due to non-cardiovascular causes, particularly GI-; GU-; and endocrine, nutrition, and metabolic-related issues. We also identified several characteristics associated with increased hospitalization, including a relatively preserved eGFR ($<60$ ml/min/1.73m$^2$), a blood pressure level of SBP of $>130$ mmHg, and the presence of moderate-to-heavy proteinuria. Hospitalization rates in the CRIC cohort were consistently higher among individuals younger than 65 years than among similarly aged individuals in the NIS, a representative US national sample of hospitalizations. These findings shed new light on how the patterns of and factors related to hospitalizations in the setting of CKD reflect the high burden of diverse morbidities in this population.

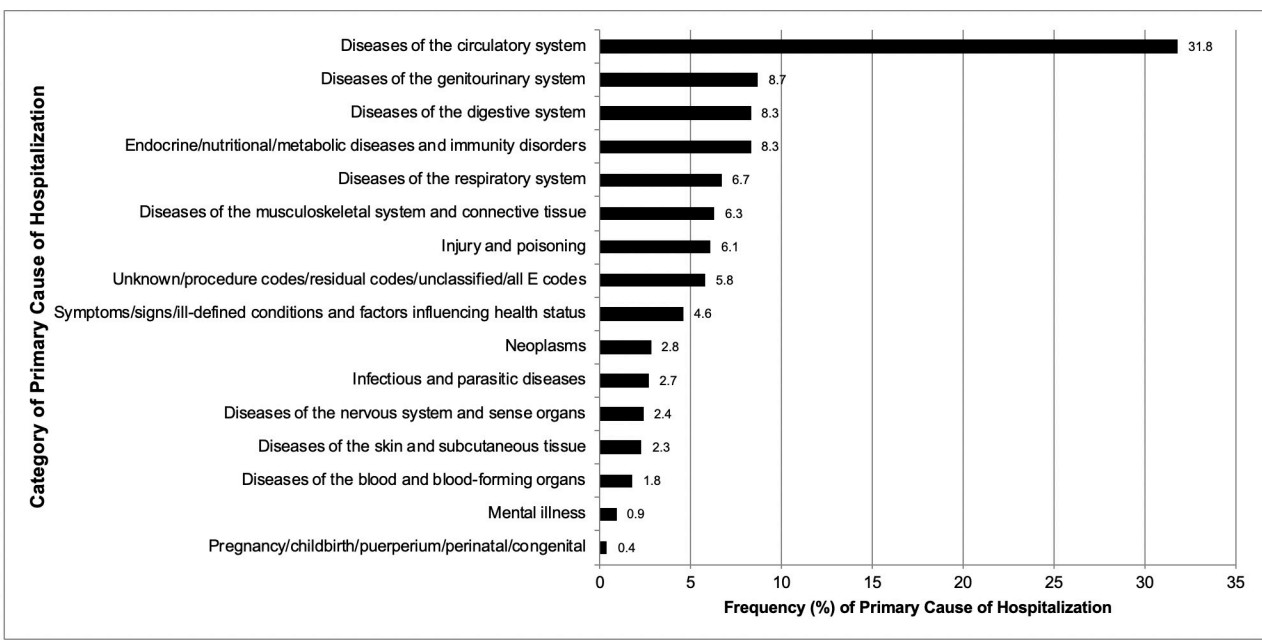

**Fig 4. Prevalence of primary cause of each hospitalization in the CRIC Study follow-up period.** CRIC, Chronic Renal Insufficiency Cohort.

Previous studies have reported that older age, female sex, concomitant cardiovascular disease, reduced eGFR, and diabetes may predict hospitalization in patients with CKD [6,7,21,23]. However, many of these previous reports were limited to retrospective cohort studies of individuals with advanced CKD, including individuals with incident ESKD. Similar to previous studies, we observed that older age and diabetes had significant associations with hospitalizations, but further extended these findings to individuals with earlier stages of CKD and to the characterization of types of hospitalization, including all-cause, cardiovascular, and non-cardiovascular hospitalizations.

The eGFR level at which hospitalization rates increase in the setting of CKD is not well characterized. In a community-based cohort study of elderly adults, the eGFR level associated with more hospitalizations was reported to be less than 30 ml/min/1.73m$^2$, but this study was limited to adults over 75 years of age [11]. In the current study, comprising a much younger cohort, hospitalization rates were higher at even at moderate reductions in kidney function (eGFR level of <60 ml/min/1.73m$^2$), and consistent with previous reports [4], we observed that individuals with eGFR of 30 to <45 ml/min/1.73m$^2$ (CKD stage 3b) had higher rates of cardiovascular-related hospitalizations than those with CKD stage 3a (eGFR of 45 to <60 ml/min/1.73m$^2$), even after adjusting for proteinuria and other comorbidities. Together, these findings suggest that patients with stage 3 CKD, who represent the majority of North Americans with CKD [1], are at an increased risk for hospitalizations. Since fewer than 10% of individuals with eGFR of <60 ml/min/1.73m$^2$ in the US are under the care of a nephrologist, awareness should be raised among primary healthcare providers of the increased hospitalization risk among this group of individuals [33,34].

The impact of proteinuria on hospitalization risk has not been well described as many of the previous studies in kidney disease populations did not account for proteinuria [6,8,10,19,21,35–38]. However, in 1 large community-based longitudinal study of patients with CKD, Hemmelgarn and colleagues reported that the risks of hospitalization for myocardial infarction independently increased in patients with higher levels of proteinuria at a given level

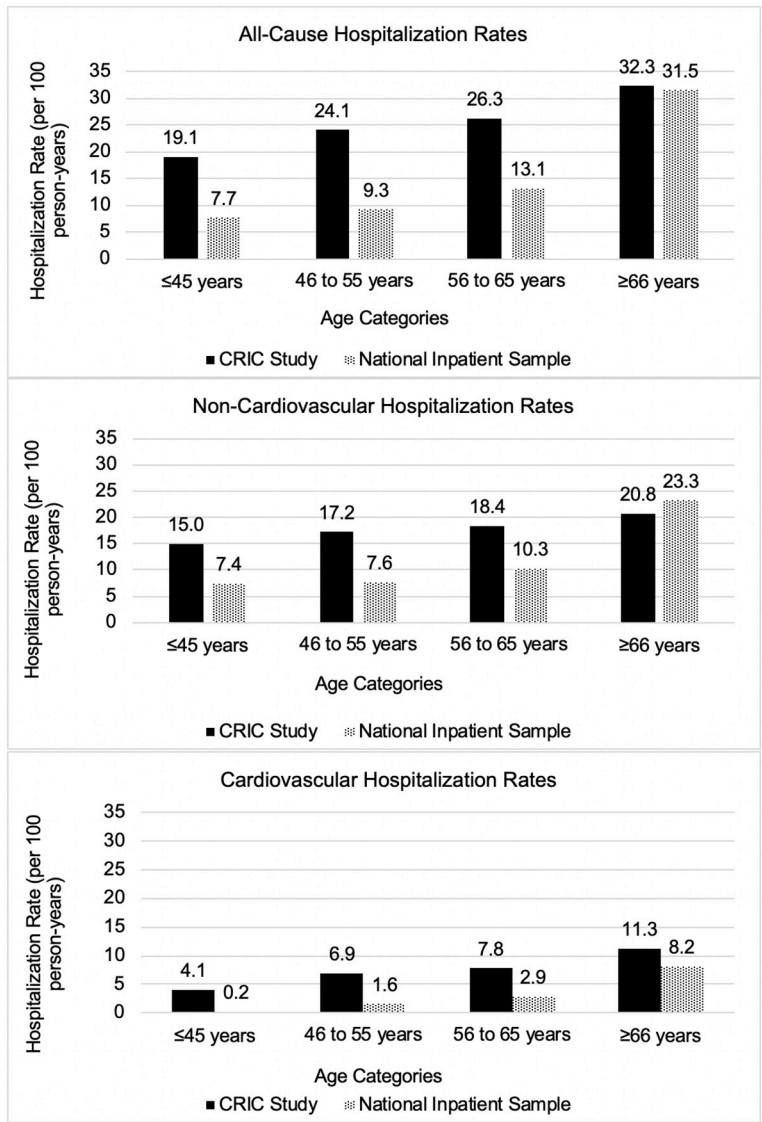

**Fig 5. Hospitalization rates (per 100 person-years) across age categories in the CRIC Study and national inpatient sample.** CRIC, Chronic Renal Insufficiency Cohort.

of eGFR [39]. Extending these findings, we observed that participants with the heaviest proteinuria experienced higher rates of all types of hospitalization across a wide range of eGFR levels, similar to the rates observed among participants with eGFR of <30 ml/min/1.73m$^2$ at any proteinuria level. Even in the presence of moderate proteinuria (150 to <500 mg/g), hospitalization rates were particularly high among those with eGFR of <45 ml/min/1.73m$^2$. Together, these findings support the need to characterize the risk of hospitalization based on both eGFR and severity of proteinuria, which reinforces the utility of the Kidney Disease Improving Global Outcomes (KDIGO) staging system for CKD, which accounts for both eGFR and proteinuria [40]. Doing so could more effectively identify individuals at highest risk for being hospitalized.

We observed that women with mild-to-moderate CKD experienced more all-cause and non-cardiovascular hospitalizations but fewer cardiovascular-related hospitalizations than

men. Potentially, these observations may be related to a greater rate and severity of complications of kidney disease in women, leading to increased hospitalizations for non-cardiovascular causes, such as electrolyte disorders or underestimation of kidney function, compared with men [7]. Additionally, there may be differences in the prevention, recognition, and care of cardiovascular disease between men and women. Previous literature has reported that physicians are less likely to discuss with women cardiovascular disease risk and provide quality preventative care or refer for cardiac catheterization when presenting with chest pain compared to men, and historically, women have been excluded from cardiovascular clinical trials [41–43]. Additionally, participants who identified as NH black or Hispanic experienced more cardiovascular hospitalizations than NH white participants. This finding could be attributed to the rising rate of cardiovascular disease and its risk factors among these minority groups compared to NH whites [44].

Individuals with CKD have been reported to be hospitalized more than twice as frequently in the preceding year as individuals without CKD [9]. In the current study, we expanded on these findings and report actual rates of hospitalization in a diverse cohort of individuals with known CKD, comparing with rates in a nationally representative sample of all hospitalized Americans. We observed that the rates for all-cause and cardiovascular-related hospitalizations were approximately 2-fold higher in CRIC than in the NIS, while non-cardiovascular hospitalization rates among the group older than 65 years were similar. These findings further demonstrate that individuals with CKD are hospitalized for a diverse set of causes more frequently at all ages. They further highlight the higher rate hospitalization in the setting of CKD compared with the rest of the US population.

In the literature, the main burden of hospitalizations in CKD has been attributed to cardiovascular causes, estimated to be 21% to 32% of all hospitalizations [3,10,20]. Several authors have also reported that after cardiovascular causes, the next largest contributor of hospitalizations is due either to infections (22%) [3] or digestive diseases (9% to 11%) [10,20], which leaves uncharacterized the majority of hospitalizations that occur in the CKD population not yet characterized. In the current study, we also identified that cardiovascular-related and digestive/GI-related causes were responsible for a large amount of hospitalizations (40.1%) occurring during the CRIC follow-up. This study has also pinpointed 2 other large contributors to the burden of hospitalization that have not yet been described in the CKD population, namely, GU-related causes and endocrine, nutrition, metabolic-related causes. The GU-related CCS category includes conditions related to kidney disease itself, such as acute kidney injury and chronic renal failure, as well as disorders of the urinary tract, including hydronephrosis, calculi, and infections (refer to S1 Text for more details). The broad category of endocrine, nutrition, and metabolic-related hospitalizations includes not only many categories clearly related to endocrine and nutrition issues, such as diabetes, thyroid disorders, and nutritional deficiencies, but also causes that could be attributed to kidney disease, including gout, fluid and electrolyte disorders, and disorders of mineral metabolism. These hospitalizations were particularly common among CRIC participants with diabetes compared with those without diabetes (11.6% versus 4.2%, respectively), which is especially important since almost half of individuals with CKD have diabetes [45]. Overall, close to 70% of hospitalizations in CKD are not due to cardiovascular disease, which is often thought of as the principal morbidity in CKD, and our findings support that recognizing CKD as a risk factor for acute kidney injury, and providing guideline-based management for CKD care could be a very important strategy to reduce hospital admissions among those with CKD. Future efforts directed at the underlying causes of endocrine, nutrition, and metabolic-related admissions, such as managing and/or preventing electrolyte abnormalities, polypharmacy, and ensuring correct medication dose adjustments, could further reduce these hospitalizations.

Our study has several potential limitations that should be considered when interpreting its findings. First, diagnostic codes and the CCS categorizations are subject to misclassification. We suspect that the low number of hospitalizations attributed to infections is due to the CCS defined categories, since infections could also be categorized into individual organ systems in addition to the infection category. Given the large number of hospitalizations across various regions of the US, however, we expect this to not invalidate our aggregate results. Also, as with all observational studies, residual confounding is possible. Additionally, the CRIC Study enrolled volunteers, which may limit generalizability of these findings. In particular, enrollment criteria excluded those with advanced heart disease, which may have biased our results toward more non-cardiovascular hospitalizations. Our study also has several important strengths including the large prospective cohort of well-characterized participants with mild-to-moderate CKD from 7 clinical centers across the US whose hospitalizations are captured through multiple approaches of ascertainment. Further, we were able to adjust hospitalization rates for proteinuria, an important outcome prediction marker, which has been a prior limitation of many previous studies related to hospitalization in the CKD population. Additionally, the NIS is estimated to represent approximately 95% of all hospital discharges in the US. Finally, the CRIC Study populations' age distribution is notably broader than was the distribution of other prospective observational cohorts with CKD that have principally examined hospitalization data in older individuals.

## Conclusion

The observed hospitalization rate among those with CKD was much higher than the rate in the general population ≤65 years of age that is hospitalized in the US, and the hospitalizations typically occurred at kidney function levels not requiring nephrology care. Although cardiovascular illness is the most common cause of hospitalization, a majority of hospital stays are due to non-cardiovascular illnesses, including GI-; GU-; and endocrine, nutrition, and metabolic-related causes. Heavy proteinuria is a potent risk factor for hospitalization. These findings highlight the need for developing better approaches to identifying patients at risk for severe complications of CKD and to developing mitigation strategies to improve outcomes in CKD.

## Supporting information

**S1 Fig. Curves from model-based associations of age, eGFR, UPCR, and SBP modeled continuously with hospitalization rates.**
(TIF)

**S2 Fig. The prevalence of the primary cause of each hospitalization in CRIC by diabetes status.**
(TIF)

**S3 Fig. The prevalence of the primary cause of each ≤1-day hospitalization in CRIC by diabetes status.**
(TIF)

**S1 Table. Modified STROBE Statement.**
(DOCX)

**S2 Table. Unadjusted all-cause hospitalization rates.**
(DOCX)

**S3 Table. Age-, race-, and diabetes-adjusted all-cause, cardiovascular-, and non-cardiovascular-related hospitalization rates.**
(DOCX)

**S4 Table. Multivariable-adjusted associations of demographic and kidney-related characteristics and hospitalization rates.**
(DOCX)

**S5 Table. Multivariable-adjusted all-cause, cardiovascular, and non-cardiovascular hospitalization rates in CRIC participants by proteinuria and eGFR level during the follow-up period.**
(DOCX)

**S6 Table. Unadjusted rate of all-cause, cardiovascular, and non-cardiovascular ≤1-day hospitalizations by key baseline characteristics of CRIC participants (*N* = 3,939).**
(DOCX)

**S7 Table. Multivariable adjusted rates of all-cause, cardiovascular, and non-cardiovascular ≤1-day hospitalizations by age, race/ethnicity, and diabetes of CRIC participants (*N* = 3,939).** Rates reported as per 100 person-years.
(DOCX)

**S8 Table. Multivariable adjusted rate ratios of all-cause, cardiovascular, and non-cardiovascular ≤1-day hospitalizations by key baseline characteristics of CRIC participants (*N* = 3,939).**
(DOCX)

**S9 Table. Multivariable-adjusted all-cause, cardiovascular, and non-cardiovascular ≤1-day hospitalization rates in CRIC participants by proteinuria and eGFR level during the follow-up period.**
(DOCX)

**S1 Text. Categories of CCS.**
(DOCX)

**S2 Text. Analytical Planning Document.**
(DOCX)

## Acknowledgments

The authors thank the CRIC Study investigators (Lawrence J. Appel, MD, MPH; Alan S. Go, MD; Jiang He, MD, PhD; Robert G. Nelson, MD, PhD, MS; Panduranga S. Rao, MD; Mahboob Rahman, MD; Vallabh O. Shah, PhD, MS; Raymond R. Townsend, MD; and Mark L. Unruh, MD, MS), staff, and participants for their important contributions.

## Author Contributions

**Conceptualization:** Sarah J. Schrauben, Eugene Lin, Harold I. Feldman.

**Data curation:** James P. Lash, Jeffrey C. Fink, Mahboob Rahman, Harold I. Feldman, Amanda H. Anderson.

**Formal analysis:** Sarah J. Schrauben, Hsiang-Yu Chen, Wei Yang.

**Funding acquisition:** James P. Lash, Jeffrey C. Fink, Mahboob Rahman, Harold I. Feldman, Amanda H. Anderson.

**Investigation:** Eugene Lin, Christopher Jepson, Julia J. Scialla, Michael J. Fischer, James P. Lash, Jeffrey C. Fink, L. Lee Hamm, Radhika Kanthety, Mahboob Rahman, Harold I. Feldman, Amanda H. Anderson.

**Methodology:** Sarah J. Schrauben, Hsiang-Yu Chen, Eugene Lin, Wei Yang, Harold I. Feldman, Amanda H. Anderson.

**Project administration:** Harold I. Feldman.

**Resources:** Harold I. Feldman.

**Supervision:** Harold I. Feldman, Amanda H. Anderson.

**Writing – original draft:** Sarah J. Schrauben, Harold I. Feldman, Amanda H. Anderson.

**Writing – review & editing:** Sarah J. Schrauben, Hsiang-Yu Chen, Eugene Lin, Christopher Jepson, Wei Yang, Julia J. Scialla, Michael J. Fischer, James P. Lash, Jeffrey C. Fink, L. Lee Hamm, Radhika Kanthety, Mahboob Rahman, Harold I. Feldman, Amanda H. Anderson.

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
