## [Editor Report · Decision Letter 0]

11 May 2020

Dear Dr Schrauben, 

Thank you for submitting your manuscript entitled "Hospitalization Experience of Adults with Chronic Kidney Disease: Findings from the Chronic Renal Insufficiency Cohort (CRIC) Study" for consideration by PLOS Medicine.

Your manuscript has now been evaluated by the PLOS Medicine editorial staff and I am writing to let you know that we would like to send your submission out for external peer review.

Kind regards,

Helen Howard, for Clare Stone PhD 

Acting Editor-in-Chief

PLOS Medicine 

plosmedicine.org

---

## [Decision Letter · Decision Letter 1]

18 Jun 2020

Dear Dr. Schrauben,

Thank you very much for submitting your manuscript "Hospitalization Experience of Adults with Chronic Kidney Disease: Findings from the Chronic Renal Insufficiency Cohort (CRIC) Study" (PMEDICINE-D-20-01995R1) for consideration at PLOS Medicine. 

[LINK]

In light of these reviews, I am afraid that we will not be able to accept the manuscript for publication in the journal in its current form, but we would like to consider a revised version that addresses the reviewers' and editors' comments. Obviously we cannot make any decision about publication until we have seen the revised manuscript and your response, and we plan to seek re-review by one or more of the reviewers. 

We expect to receive your revised manuscript by Jul 09 2020 11:59PM. Please email us (plosmedicine@plos.org) if you have any questions or concerns.

We look forward to receiving your revised manuscript. 

Sincerely,

Adya Misra, PhD

Senior Editor 

PLOS Medicine

plosmedicine.org

Please revise your title according to PLOS Medicine's style. Your title must be nondeclarative and not a question. It should begin with main concept if possible. "Effect of" should be used only if causality can be inferred, i.e., for an RCT. Please place the study design ("A randomized controlled trial," "A retrospective study," "A modelling study," etc.) in the subtitle (ie, after a colon).

* Please structure your abstract using the PLOS Medicine headings (Background, Methods and Findings, Conclusions).

* Please combine the Methods and Findings sections into one section, “Methods and findings”.

Abstract Methods and Findings:

* Please ensure that all numbers presented in the abstract are present and identical to numbers presented in the main manuscript text.

* Please include the study design, population and setting, number of participants, years during which the study took place, length of follow up, and main outcome measures.

* Please quantify the main results (with 95% CIs and p values).

* Please include the important dependent variables that are adjusted for in the analyses.

Abstract Conclusions:

* Please address the study implications without overreaching what can be concluded from the data; the phrase "In this study, we observed ..." may be useful.

* Please interpret the study based on the results presented in the abstract, emphasizing what is new without overstating your conclusions.

* Please avoid vague statements such as "these results have major implications for policy/clinical care". Mention only specific implications substantiated by the results.

* Please avoid assertions of primacy ("We report for the first time....")

References- please use square brackets throughout and use Vancouver style for bibliography 

Methods- please briefly state where the participants were recruited in the US

On page 8- please remove the information regarding the funding source as this is pulled from the article meta-data only

Please provide 95% confidence intervals and p values throughout as needed. Please note we need exact p values unless p<0.001

Please present and organize the Discussion as follows: a short, clear summary of the article's findings; what the study adds to existing research and where and why the results may differ from previous research; strengths and limitations of the study; implications and next steps for research, clinical practice, and/or public policy; one-paragraph conclusion.

Did your study have a prospective protocol or analysis plan? Please state this (either way) early in the Methods section.

STROBE- please specify where individual items on the checklist can be found using paragraphs and sections. 

The Data Availability Statement (DAS) requires revision. For each data source used in your study: 

Comments from the reviewers:

Reviewer #1: I confine my remarks to statistical aspects of this paper. The general approach is fine but I have a couple issues to resolve before i can recommend publication.

The biggest one is that continuous independent variables should not be categorized. In *Regression Modeling Strategies* Frank Harrell lists 11 problems that this causes and sums up "nothing could be more disastrous". I wrote a blog post taking a graphical look at what can happen: https://medium.com/@peterflom/what-happens-when-we-categorize-an-independent-variable-in-regression-77d4c5862b6c?source=friends_link&sk=1428cd15968e218268121dc507ce8025

Other issues: p. 7 Please operationalize all variables. 

p 9 Give the SD for age

Fig 1: Use a line plot for age and do not use "dynamite plots" (see e.g http://biostat.mc.vanderbilt.edu/wiki/Main/DynamitePlots)

Fig 3: Do not use 3 dimensional plots. William S. Cleveland has shown that these cannot be accurately read. But the variables should be continuous, so this plot will change a lot.

Peter Flom

Reviewer #2: Schrauben et al. explored hospitalization rates among persons with CKD in the CRIC observational study and contrasted with hospitalization rates in the Nationwide Inpatient Sample (NIS) for 2012. They found higher hospitalization rate in the CRIS than NIS, predominantly related to diverse illnesses other than cardiovascular disease.

I offer the following suggestions for consideration:

1. The concluding statement in the abstract "Adults with CKD appear to have a substantially higher hospitalization rate than those without CKD" is not supported by the results. 

2. On page 8, under statistical analysis, please specify the start of follow-up in the CRIC cohort and the median follow-up time. Also, describe how the hospitalization rates were calculated for the NIS.

3. Please indicate the reasoning for selecting the year 2012 for NIS data.

4. NIS data represent the number of discharge diagnoses per hospital stay, not per patient, so the incidence of any hospitalization is overestimated, an aspect that should be discussed as a limitation. Along this line, it is not clear if the numerator for the CRIC data included the first (incident) hospitalization during the follow-up period, or any hospitalizations. If the former is true, the authors may consider discussing how this aspect affects the comparison with NIS rates. The authors ought to specify how the numerator was counted in each of the two data sources.

Reviewer #3: This is an interesting paper and highlights a major issue for patients with even fairly mild chronic kidney disease. The data presented genuinely contribute to novel understanding of this issue and in my opinion will be useful for research, healthcare policy and resource allocation. The CRIC cohort is well known, but this looks like a completely new and more detailed analysis compared to any of their other work. 

To my reading, this is well presented, the analyses look sound and the tables and figures are informative. I have little to suggest to this and I think this could be a very useful paper for others working in this field.

My main comments are around the grammar, or messaging in the paper

Abstract.

1.I presume they mean 500mg/g of proteinuria? - there is no explanation as to what this measurement refers to. Also why present this threshold in the abstract when most of the thresholds for uPCR in the paper are 150mg/g. Is this correct? I presume this is as 500mg/g is the upper category - it could be clarified as to why this value appears where it does

2. Although I accept that the majority of illnesses are not cardiovascular, the single biggest disease by far affecting these patients is cardiovascular disease as their analysis shows (31.8% compared to 8.7% for the next condition down). I think the message should be slightly rephrased to make sure that the message about CVD is not diminished too much

Main text

They should go through this carefully for units etc. I found a few instances where the units had dropped off inappropriately e.g. for GFR on p10. There are probably others. I don't view it as the role of reviewer to be a proof reader!

[LINK]

---

## [Decision Letter · Decision Letter 2]

26 Oct 2020

Dear Dr. Schrauben,

Thank you very much for re-submitting your manuscript "Description of Hospitalizations Among Adults with Chronic Kidney Disease: An Observational Longitudinal Cohort Study" (PMEDICINE-D-20-01995R2) for review by PLOS Medicine.

I have discussed the paper with my colleagues and the academic editor and it was also seen again by xxx reviewers. I am pleased to say that provided the remaining editorial and production issues are dealt with we are planning to accept the paper for publication in the journal.

[LINK]

We look forward to receiving the revised manuscript by Nov 02 2020 11:59PM. 

Sincerely,

Adya Misra, PhD

Senior Editor 

PLOS Medicine

plosmedicine.org

Requests from Editors:

Title- I suggest revising to “ Hospitalizations Among Adults with Chronic Kidney Disease in the United States: A Cohort Study" or similar.

Abstract

Please provide brief participant demographics and where the cohort is located

Please include 2-3 limitations of your study in the last sentence of the methods and findings section

Please revise to ""had" in the first line of the conclusions 

Throughout- please place reference brackets before punctuation, providing a space in between.

Please provide p-values up to three decimal places throughout.

Page 16 first paragraph third sentence contains an additional “to”

STROBE checklist- please remove page numbers as these are likely to change during publication. Please use paragraphs and sections instead. The STROBE and protocol documents can be uploaded as separate attachments, referred to in the methods

Please provide access details for ref 19, 36. Reference 1 needs 6 author names (+ page numbers?)

At the end of the ms, the acknowledgements are mainly funding, I think, which is in the submission form and can be removed. 

Please ensure all p-values provided are up to three decimal places. All iterations of p<0.0001 should be changed to p<0.001

Comments from Reviewers:

Reviewer #1: The authors have addressed my concerns and I now recommend publication

Peter Flom

Reviewer #2: The authors' responses are satisfactory and i have no further comments. Thank you.

Reviewer #3: I have no further comments

[LINK]

---

## [Editor Report · Decision Letter 3]

5 Nov 2020

Dear Dr. Schrauben, 

On behalf of my colleagues and the academic editor, Dr. Meda E Pavkov, I am delighted to inform you that your manuscript entitled "Hospitalizations Among Adults with Chronic Kidney Disease in the United States: A Cohort Study" (PMEDICINE-D-20-01995R3) has been accepted for publication in PLOS Medicine. 

PRODUCTION PROCESS

Before publication you will see the copyedited word document (within 5 business days) and a PDF proof shortly after that. The copyeditor will be in touch shortly before sending you the copyedited Word document. We will make some revisions at copyediting stage to conform to our general style, and for clarification. When you receive this version you should check and revise it very carefully, including figures, tables, references, and supporting information, because corrections at the next stage (proofs) will be strictly limited to (1) errors in author names or affiliations, (2) errors of scientific fact that would cause misunderstandings to readers, and (3) printer's (introduced) errors. Please return the copyedited file within 2 business days in order to ensure timely delivery of the PDF proof. 

If you are likely to be away when either this document or the proof is sent, please ensure we have contact information of a second person, as we will need you to respond quickly at each point. Given the disruptions resulting from the ongoing COVID-19 pandemic, there may be delays in the production process. We apologise in advance for any inconvenience caused and will do our best to minimize impact as far as possible.

PRESS

PROFILE INFORMATION

Thank you again for submitting the manuscript to PLOS Medicine. We look forward to publishing it. 

Best wishes, 

Adya Misra, PhD

Senior Editor 

PLOS Medicine

plosmedicine.org